# Salvage Chemotherapy in Patients with Previously Treated Thymic Carcinoma

**DOI:** 10.3390/cancers13215441

**Published:** 2021-10-29

**Authors:** Kyoichi Kaira, Hisao Imai, Ou Yamaguchi, Atsuto Mouri, Hiroshi Kagamu

**Affiliations:** Department of Respiratory Medicine, Comprehensive Cancer Center, International Medical Center, Saitama Medical University, 1397-1 Yamane, Hidaka 350-1298, Saitama, Japan; m06701014@gunma-u.ac.jp (H.I.); ouyamagu@saitama-med.ac.jp (O.Y.); mouria@saitama-med.ac.jp (A.M.); kagamu19@saitama-med.ac.jp (H.K.)

**Keywords:** thymic carcinoma, salvage chemotherapy, molecular targeting agent, platinum, refractory

## Abstract

**Simple Summary:**

Thymic carcinoma is identified as thoracic neoplasm having low sensitivity to systemic chemotherapy. As first-line setting, platinum-based chemotherapy is administered, but, it is difficult to achieve long-term survival. Therefore, salvage chemotherapy is clinically considered as second or third line treatment. This study reviewed the therapeutic significance of several kinds of cytotoxic agents and molecular targeting drugs in patients with previously treated thymic carcinoma. The clinical trials of salvage chemotherapy in patients with thymic carcinoma are limited, and we cannot draw an optimal conclusion due to the small sample size. However, S-1, amrubicin, docetaxel, pemetrexed, and paclitaxel, sunitinib and lenvatinib yielded some efficacy to such patients. As S-1 and amrubicin are limited to Japan and East Asian, it remains unclear which regimens are better as second-line setting. Further investigation is warranted to establish the clinical evidence of salvage chemotherapy in advanced or metastatic thymic carcinoma by large-scale study.

**Abstract:**

Thymic carcinoma is a rare neoplasm, and it is difficult to achieve complete remission with systemic chemotherapy. In advanced or recurrent thymic carcinoma, platinum-based chemotherapy is chosen as the first-line setting; however, it remains unclear which regimen is better to improve its outcome. It remains unknown whether salvage chemotherapy should be administered to patients with platinum-based chemotherapy-refractory thymic carcinoma. Currently, several clinical studies have investigated the efficacy of second-line settings for advanced thymic carcinoma. As cytotoxic agents, S-1, amrubicin, pemetrexed, docetaxel, paclitaxel, and gemcitabine have been reported as prospective phase II studies or retrospective studies. The overall response rates (ORRs) of S-1, amrubicin, and pemetrexed were 25–50%, 11–44.4%, and 9–10%, respectively. Molecular targeting drugs, such as sunitinib, everolimus, and lenvatinib, also provide clinical effectiveness with tolerability after the failure of platinum-based regimens. Based on the results of the prospective phase II study, the ORR, median progression-free survival, and median overall survival were 16.6% and 5.6 months, respectively, in everolimus, 26% and 7.2 months, respectively, in sunitinib, and 38% and 9.3 months, respectively, in lenvatinib. Although it is difficult to compare each study, lenvatinib appears to be better in increasing efficacy as a second-line setting. However, each study had a small sample size, which may have biased the results of their studies. Further investigation is warranted to elucidate the therapeutic significance of salvage chemotherapy in advanced thymic carcinoma in a large-scale study.

## 1. Introduction

Thymic epithelial tumors consist of thymoma and thymic carcinoma (TC), which are rare neoplasms. Notably, TC is a dismal entity that easily depicts recurrence after surgical resection. In patients with metastatic or advanced TC, systemic chemotherapy is widely chosen to decrease tumor growth and metastasis. but the tumor is frequently resistant to chemotherapy. To date, a combination of platinum-based regimens is administered to patients with TC as a first-line setting; however, it remains debatable which regimens are appropriate for such patients in terms of efficacy and survival. Despite no formal evidence of survival benefits with salvage chemotherapy, its use is widely supported by clinical practice and high disease control rate. However, many patients are treated with salvage chemotherapy.

A recent review on systemic chemotherapy included patients with thymoma and TC together [1]. Cisplatin plus anthracycline or cisplatin with etoposide combination is recommended as first-line chemotherapy for thymoma and TC, although anthracycline is a key drug for thymoma, not the same for TC [1]. A recent phase II study demonstrated that carboplatin and paclitaxel yielded moderate clinical activity and less toxicity in patients with thymic malignancies compared with anthracycline-based regimens [2]. Considering the balance between toxicity and efficacy, carboplatin plus paclitaxel may be better in clinical practice. Meanwhile, some cytotoxic agents have been chosen for patients with TC after failure of platinum-based first-line chemotherapy; however, the content of their regimens depends on the individual institutional protocol. We have not answered which regimens are appropriate for salvage chemotherapy. Meanwhile, several studies have reported the efficacy and tolerability of different types of cytotoxic agents using small sample sizes, and using a prospective or retrospective approach.

Recent oncologic topics include immune checkpoint inhibitors (ICIs), such as programmed death-1 (PD-1)/programmed death ligand-1 (PD-L1), which are widely administered to several types of human neoplasms and encourage them to improve outcomes after treatment. We have previously reviewed the perspectives of immunotherapy in patients with TC and discussed their potential therapeutic efficacy [3]. Although the usefulness of ICIs is expected to result in long-term survival, salvage chemotherapy with cytotoxic agents may also be essential for the sequential treatment of advanced or metastatic patients with TC. Therefore, we reviewed the clinical evidence for salvage chemotherapy in addition to ICIs and discussed the clinical usefulness of sequential treatment in such patients.

## 2. Clinical Evidence of First-Line Chemotherapy in TC

As appropriate chemotherapeutic regimens in the first-line setting, platinum-based chemotherapy has shown some active efficacy and tolerability in patients with thymic epithelial tumors, including thymoma and TC. A recent review described that combinations of cisplatin-anthracycline or cisplatin-etoposide are recommended as first-line chemotherapy in such patients [2]. Table 1 shows a summary of platinum-based chemotherapy as first-line treatment in patients with advanced TC. Previous studies have identified that cisplatin-adriamycin-cyclophosphamide (CAP), cisplatin-doxorubicin-cyclophoshamide-vincristine (ADOC), cisplatin-etoposide (VP16), cisplatin-docetaxel, and carboplatin-paclitaxel are the most common regimens administered in patients with TC [2,4,5,6,7,8,9,10,11,12,13,14,15]. The overall response rate (ORR) of platinum-based chemotherapy was yielded approximately 30–40% (from 21% to 70%), regardless of the small sample size. Although high-intensity regimens, such as CAP or ADOC, increase the response rate, severe adverse events were also observed, although CAP is a standard regimen at least in Europe for thymoma [4,5,6,7,8,9,10]. Meanwhile, cisplatin plus etoposide or carboplatin plus paclitaxel are commonly administered to patients with advanced non-small-cell lung cancer (NSCLC) and small-cell lung cancer (SCLC), known as active and tolerable regimens. Unfortunately, multidrug chemotherapy such as ADOC or CAP has been clarified as a toxic regimen, regardless of its high efficacy. Nowadays, the useful frequency of these heavy regimens gradually decreases due to the lack of significant improvement in the outcome compared to cisplatin plus etoposide or carboplatin plus paclitaxel. Considering the results of previous studies (Table 1), the number of patients with TC registered in individual studies showed small sample sizes; therefore, it remains unclear which regimen is better as a standard treatment. However, the histological type of patients with TC predominantly consists of squamous cell carcinoma, similar to that of NSCLC. Recently, Ko et al. retrospectively analyzed the prognostic factors and efficacy of first-line chemotherapy in 286 patients with advanced TC in a multi-institutional study [16]. In their study, the administration frequency of platinum-based doubles, monotherapy, and other multidrug chemotherapy such as ADOC was 62.2%, 3.5%, and 34.3%, respectively, and there was no significant difference in overall survival (OS) between different first-line therapeutic regimens (between carboplatin plus paclitaxel and ADOC, median OS: 27.8 vs. 29.9 months) [16]. Of the 286 patients, carboplatin plus paclitaxel was administered to 70 patients with an ORR of 49%, cisplatin plus etoposide in 35 patients (ORR, 48.6%), cisplatin plus irinotecan in 16 patients (ORR, 66.7%), carboplatin plus etoposide in 15 patients (ORR, 30.8%), and cisplatin plus docetaxel in nine patients (ORR, 22.2%). As with other multidrug chemotherapy, the 79 patients who received ADOC achieved ORR of 41% and CAP were administered to eight patients with an ORR of 37.5%. They concluded that the efficacy of individual first-line regimens against advanced TC was not significantly different, and the use of carboplatin plus paclitaxel might be adequate as first-line chemotherapy [16]. In terms of efficacy, tolerability, and histological similarity, carboplatin plus paclitaxel seems appropriate as first-line chemotherapy for patients with TC.

## 3. Cytotoxic Agents as Salvage Chemotherapy

Due to their rarity, there are limited reports on salvage chemotherapy for patients with previously treated advanced TC [22,23,24,25,26,27,28,29,30,31,32,33,34,35,36,37,38,39,40,41,42,43,44,45,46,47,48,49,50,51]. Owing to the limited number of studies and small sample sizes, it is difficult to determine the most effective therapeutic regimen for such patients. Of several chemotherapeutic regimens, S-1, amrubicin, docetaxel, paclitaxel, gemcitabine, etoposide, sunitinib, everolimus, and lenvatinib have been reported as active drugs in previously treated patients with TC. Recently, Tateishi et al. retrospectively analyzed the clinical outcomes of 191 patients with previously treated advanced TC [27]. As second-line chemotherapy in their study, platinum-based doublets, multidrug chemotherapy (e.g., ADOC), and monotherapy were observed in 57.6%, 13.6%, and 28.8%, respectively. The median OS at the initiation of second-line chemotherapy was 22.4 months, and the average ORR was 20.0% (ORR and median OS were 21.6% and 22.4 months, respectively, in platinum-based doublet chemotherapy, 13.6% and 25.7 months, respectively, in other multidrug chemotherapy and 19.6% and 21.4 months, respectively, in monotherapy). The results of their study indicated that there was no significant difference in ORR and OS between each regimen. Therefore, compared to combination chemotherapy, monotherapy could be easily and safely administered to patients with advanced TC as a second-line treatment. Here, we focus on the clinical usefulness of active single-agent candidates for second-line chemotherapy.

### 3.1. S-1

S-1 (Taiho Pharmaceutical Co., Ltd., Tokyo, Japan) is an oral anticancer agent comprising tegafur (FU), 5-chloro-2,4-dihydroxypyridine (CDHP), and potassium oxonate (Oxo), in a molar ratio of 1:0.4:1 [28]. Tegafur, a 5-fluorouracil (5-FU) prodrug, is gradually converted to 5-FU and rapidly catabolized by dihydropyrimidine dehydrogenase (DPD) in the liver. In our daily practice, S-1 is an active anticancer drug in different types of human neoplasms, such as NSCLC, gastrointestinal cancer, breast cancer, head and neck cancer, hepatobiliary tract cancer, and pancreatic cancer. The anticancer activity of 5-FU is closely correlated with the intratumoral expression of thymidylate synthase (TS), orotate phosphoribosyl transferase (OPRT), and DPD [29].

Since S-1 shows relatively mild toxicity, it is easy to combine other compounds with S-1, and it is useful as late-line chemotherapy. Table 2 shows the previous reports for S-1 in patients with TC as second-line or more line settings. Two prospective studies described an ORR of approximately 30%, a median progression-free survival (PFS) of approximately 5.0 months, and a median OS of approximately 2 years [22,23]. Moreover, the same or superior efficacy in a prospective study setting was observed in four retrospective studies [24,25,26,27]. As a second-line setting, S-1 appeared to improve the response and survival in patients with advanced TC, even in small-scale studies. Several authors have reported a drastic response to S-1 monotherapy in four cases with chemorefractory TC [30,31,32,33]. By immunohistochemistry, TS as the main target molecule of S-1 has previously been reported to be positively expressed in 94% of patients with TC [34]. Notably, the expression of TS, OPRT, and DPD was closely associated with the grade of malignancy in patients with thymic epithelial tumors, and these biomarkers may play a crucial role in predicting the effectiveness of 5-FU based regimens in thymic malignancies [34]. Therefore, S-1 may depict tumor shrinkage by targeting the expression of TS within tumor cells in patients with TC.

### 3.2. Amrubicin

Amrubicin hydrochloride, a completely synthetic 9-amino-anthracycline, is converted to the active metabolite, amrubicinol, which reduces its C-13 ketone group to a hydroxyl group by carbonyl reductase [35]. Amrubicin and amrubicinol are inhibitors of DNA topoisomerase II, and amrubicinol is 5–100 times more active than amrubicin [36]. Amrubicin has been widely administered to patients with previously treated SCLC and NSCLC in Japan. Recently, several researchers have reported prospective and retrospective analyses to assess the efficacy of amrubicin in advanced patients with TC as second-line settings (Table 3) [37,38,39,40]. Due to small-scale investigations, it is difficult to decide whether amrubicin monotherapy or platinum doublets with amrubicin could yield better efficacy with tolerability. Regarding first-line chemotherapy in thymic malignancies, the anthracycline doxorubicin is a key drug in their treatment and can enhance tumor shrinkage [16]. Amrubicin is a new anthracycline as a topoisomerase II inhibitor with less cardiac toxicity than doxorubicin [9,10,11,12]. Although the histology of TC is predominantly squamous cell carcinoma, similar to NSCLC, patients with TC with small cell carcinoma are rarely observed. Igawa et al. reported a case that showed a marked response to amrubicin in a patient with TC with histology of small cell carcinoma [41]. Moreover, we have previously reported two cases in which amrubicin monotherapy could continue to control tumor progression over a long-term period [42].

### 3.3. Pemetrexed

Pemetrexed (Alimta, Eli Lilly, Kobe, Japan) is a multitargeted antifolate drug with enhanced activity in advanced NSCLC [43]. Notably, pemetrexed exhibited significant efficacy alone or in combination with platinum-based regimens in NSCLC and malignant pleural mesothelioma. Similar to the targeting mechanism of S-1, TS is also identified as a key enzyme targeted by pemetrexed and is highly expressed in squamous cell carcinoma [44]. Based on the concept of TS expression within TC cells, the possibility of pemetrexed as a target agent was challenged, and two researchers described the efficacy and clinical outcome in patients with recurrent TC [45,46,47] (Table 4). Although the ORR of around 10% appeared to be relatively lower than that of S-1 or amrubicin, the impact on the prognosis after pemetrexed initiation was comparable to that after second-line treatment.

### 3.4. Docetaxel

Docetaxel is a standard treatment for patients with previously treated NSCLC. Two previous retrospective studies demonstrated the efficacy of docetaxel monotherapy [27,48], and another study discussed the potential of docetaxel with platinum combination and docetaxel monotherapy [49] (Table 4). Although all studies were retrospective in nature, median OS may be favorable compared to that of other cytotoxic agents, ranging from 21.7 to 24.0 months [27,48,49].

### 3.5. Paclitaxel

No studies have reported the efficacy of paclitaxel monotherapy in patients with advanced TC as a second-line treatment. Carboplatin plus paclitaxel is one of the most common combinations, regardless of the histological type of the tumor. As mentioned above, carboplatin plus paclitaxel was identified as one of the first-line chemotherapies in patients with previously untreated TC. Here, two retrospective studies showed that combination chemotherapy with carboplatin and paclitaxel was effective as a second-line or over-line chemotherapy in patients with previously treated TC [27,50] (Table 4). If this regimen was administered as a first-line treatment, cytotoxic agents with different mechanisms may be suitable. Komatsu et al. found that partial response was observed in two (66.6%) of the three cases of TC who sequentially received carboplatin plus paclitaxel as second-or third-line chemotherapy after failure of ADOC [51]. Based on case reports, nab-paclitaxel plus carboplatin has been reported to yield a clinical response in previously treated patients with TC, and carboplatin with nab-paclitaxel was effective as a third-line treatment in patients with histology of thymic large cell neuroendocrine carcinoma [52,53]. Although it depends on the condition of the patients, carboplatin plus paclitaxel may be useful not only as a first-line treatment but also as a second-line treatment.

## 4. Molecular Target Agents as Salvage Chemotherapy

Recently, molecular targeting drugs, such as vascular endothelial growth factor (VEGF) and mammalian target of rapamycin (mTOR) inhibitors, have been explored in patients with recurrent TC [54,55,56,57,58]. As these targeting agents can efficiently kill tumor cells by disrupting their signaling pathways, they may contribute to longer survival without severe toxicities. Here, several molecular targeting drugs were reviewed, and data from several clinical studies were introduced.

### 4.1. Everolimus

Everolimus is an oral mTOR inhibitor that is administered to several human neoplasms. Zucali et al. reported a phase II study to elucidate the efficacy of everolimus in thymoma and TC as a platinum-refractory case [54]. In their study, the ORR of thymoma and TC was 9.4% and 16.7%, respectively; however, the disease control rate was 93.8% in thymoma and 77.8% in TC. Overall, everolimus could prevent progressive disease (PD) in 88% of patients with thymic epithelial tumors, although the percentage of tumor shrinkage was slightly lower. Approximately half of all patients experienced PD free after 6 months of everolimus initiation. The toxicity profile of everolimus in patients with TC was consistent with that reported in previous studies. Notably, 28% of patients with thymic epithelial tumors experienced grade 1 or 2 pneumonitis, and 3 (6%) of them died of pneumonitis. Therefore, everolimus-related pneumonitis should be considered (Table 5).

### 4.2. Sunitinib

Sunitinib is an oral multi-tyrosine kinase inhibitor that targets VEGFR, KIT, and PDGFR. The tumor pathogenesis of thymic epithelial tumors has been described to be related to presence of glucose metabolism, hypoxia, and angiogenesis [59]. Angiogenesis is determined by VEGF and microvessel density (CD31 and CD34) [59]. Recently, Thomas et al. reported a phase II study of sunitinib in 40 patients with chemotherapy-refractory thymic epithelial tumors [56]. The ORR and disease control rates in TC (*n* = 23) and thymoma (*n* = 16) were 26% and 91%, respectively, and 6% and 81%, respectively [56]. Sunitinib showed limited activity in thymoma, and tumor molecular profiling exhibited somatic variations that were not predictive of response to sunitinib. Remon et al. reported that ORR and disease control rates were 20% and 55%, respectively, in TC and 29% and 86%, respectively, in thymoma [57]. However, they reported a median PFS of 5.4 months in thymoma and 3.3 months in TC, respectively; thus, the survival time was lower than that of the previous phase II study of sunitinib [56] (Table 5). Due to the limited number of registered patients, further large-scale studies are warranted to evaluate the survival benefit of sunitinib.

### 4.3. Lenvatinib

Lenvatinib is an oral multi-targeted tyrosine kinase inhibitor for VEGFR, FGFR, c-Kit, and other kinases and has been approved for use in patients with thyroid cancer and hepatocellular cancer. Similar to sunitinib, VEGF is overexpressed in TC and has been identified as a reasonable target site. Sato et al. reported a phase II study to examine the efficacy of lenvatinib in 42 patients with TC [58] (Table 5). The ORR and disease control rates were 38% and 95%, respectively. Compared to previous molecular targeting drugs, lenvatinib appeared to improve efficacy and clinical benefits in patients with TC. Although lenvatinib is approved in Japan, real-world data is necessary to accumulate patient numbers in daily practice to elucidate the survival benefit of lenvatinib in patients with TC.

## 5. Discussion

The clinical trials of salvage chemotherapy in patients with TC are limited, and we cannot draw an optimal conclusion due to the small sample size. However, some chemotherapy is clinically considered as a salvage treatment for patients with chemotherapy-refractory TC. The cytotoxic agents of salvage chemotherapy, S-1, amrubicin, docetaxel, pemetrexed, and paclitaxel, are reviewed here. Evidence of S-1 in previously treated patients with TC has been reported by all Japanese researchers. The ORR, median PFS, and median OS of S-1 ranged from 25% to 50%, 4.3 to 8.3 months, and 13.5 to 30.0 months, respectively. Although the number of registered studies was very small and there were only two prospective studies (phase II study), S-1 may be active with mild tolerability as second-line chemotherapy. Amrubicin is widely used as second-line chemotherapy for platinum-refractory SCLC in Japan. In addition to SCLC, amrubicin has been approved as salvage chemotherapy for NSCLC. In patients with TC, amrubicin monotherapy resulted in mild ORR, ranging from 11% to 44.4%. Furthermore, Inoue et al. reported that carboplatin plus amrubicin exhibited an ORR of 30%, median PFS of 7.6 months, and median OS of 27.3 months in chemotherapy-refractory patients with TC [39]. Considering the tumor shrinkage, carboplatin plus amrubicin appeared to be better than amrubicin monotherapy. Monotherapy or combination chemotherapy should be chosen according to the condition of the patient. Pemetrexed is the standard of care for patients with advanced NSCLC, especially adenocarcinoma. It is well known that pemetrexed is more active in patients with adenocarcinoma than in those with squamous cell carcinoma. The histology of TC is predominantly squamous cell carcinoma; therefore, pemetrexed may yield a lower response rate (range, 9–10%). For taxanes, such as docetaxel or paclitaxel, platinum-combination regimens (e.g., carboplatin plus paclitaxel, cisplatin plus docetaxel) are chosen for chemotherapy-refractory TC. Therefore, the ORR of these combinations was more than 25%, with increased toxicity. However, combination chemotherapy did not have a positive impact on survival.

To date, conventional chemotherapy has been discussed regarding the efficacy of salvage treatment in patients with TC. New molecular targeting agents, such as sunitinib, everolimus, and lenvatinib, are expected to improve efficacy and outcome. As described above, VEGF is highly expressed in patients with TC; therefore, inhibition of angiogenesis is identified as a reasonable target. Sunitinib provided a mild tumor shrinkage of approximately 25% in ORR compared to that of S-1; moreover, the efficacy of everolimus was lower. However, lenvatinib showed a promising efficacy of 38% in ORR and 9.3 months in median PFS, which could be one of the standard treatments for patients with TC receiving platinum-based regimens.

Currently, ICIs are the standard of care for patients with several human neoplasms. In particular, PD-1 blockade has been identified as a first-line regimen in patients with lung cancer. Several clinical studies have been proven to be effective in patients with advanced TC. We have previously reviewed the perspectives of immunotherapy in patients with TC [3]. However, we make mention of the potential of ICIs in thymic carcinoma. NIVOTHYM is a first international multicenter phase II study evaluating the usefulness of nivolumab ± ipilimumab in patients with advanced type B3 thymoma or thymic carcinoma [60]. Thymoma and thymic carcinoma were observed in 10 (18%) and 45 (82%), respectively, and ORR, PFS and OS were 12%, 6.0 months and 21.3 months, respectively. The results of PD-1 blockade monotherapy as salvage treatment in TC patients are listed in Table 6. Herein, the detailed information of immunotherapy is omitted, but the potential of ICIs is anticipated.

Based on this evidence, cytotoxic agents, such as S-1 or amrubicin, may be relatively effective for chemotherapy-refractory cases compared to other cytotoxic drugs. As S-1 or amrubicin is available in a limited number of countries, including Japan and East Asia, they are not suitable as common treatments for salvage chemotherapy. However, S-1 or amrubicin are not a key drug in Europe and USA. In addition to Japan or East Asia, pemetrexed, docetaxel, or gemcitabine can be administered according to the condition of the patients. The combination of platinum plus taxane could increase the percentage of tumor shrinkage but could also cause treatment-related toxicities compared to taxane monotherapy. If the status of the patient’s condition is better, with platinum plus taxane combination chemotherapy may be chosen as the second-line treatment. Besides, the preliminary results of capecitabine and gemcitabine (CAP-GEM) was reported in patients with advanced thymic epithelial tumors [64]. Although the ORR of this regimen was 40%, thymic carcinoma was recognized in 3 (20%) of 15 patients. Bluthgen et al. reported the efficacy of oral etoposide in 20 patients with pretreated thymic epithelial tumors, and the ORR, median PFS and median OS in 15 TC patients were observed in 13%, 4 months and 13 months, respectively [65]. Moreover, ifosfamide was examined in patients with invasive thymoma [66].

## 6. Conclusions

Among the molecular targeting agents, lenvatinib appears to be more effective than sunitinib or everolimus. Although careful monitoring is necessary to administer lenvatinib, this regimen is recommended based on current evidence. Aside from these molecular targeting agents, imatinib and dasatinib are described to be partially effective for the KIT mutated TC patients [67]. However, each study had a small sample size, which may have biased the results of their studies. Further investigation is warranted to elucidate the therapeutic significance of salvage chemotherapy in advanced TC in a large-scale study.

## Figures and Tables

**Table 1 cancers-13-05441-t001:** Reports of platinum-based regimens as first line setting in thymic carcinoma.

First Author (Year)	Ref.	Regimens	Thymoma + TC		TC	
			*N*	ORR (%)	*N*	ORR (%)
Platinum-anthracycline based chemotherapy						
Kim (2004)	[6]	CAP	22	77%	12	NA
Li (2007)	[4]	CAP	28	71%	18	61%
Cardillo (2010)	[5]	CAP	21	58%	10	50%
Agatsuma (2011)	[7]	ADOC	NA	NA	34	50%
Rea (2011)	[8]	ADOC	38	68%	6	50%
Yoh (2003)	[17]	CODE	NA	NA	12	42%
Oshita (1995)	[9]	CAP-VP16	14	43%	7	42%
Thomas (2014)	[10]	CAP-belionstat	26	40%	14	21%
Platinum-etoposide based chemotherapy						
Loehrer (2001)	[11]	CDDP+VP16-IFO	28	32%	8	NA
Grassin (2010)	[12]	CDDP+VP16-IFO	16	25%	4	25%
Platinum-taxane based chemotherapy						
Park (2013)	[13]	CDDP-DTX	27	63%	18	66%
Kim (2015)	[14]	CDDP-PTX	42	63%	28	70%
Lemma (2011)	[2]	CBDCA-PTX	44	32%	23	21.7%
Igawa (2010)	[15]	CBDCA-PTX	NA	NA	11	36%
Furugen (2011)	[18]	CBDCA-PTX	NA	NA	16	37.5%
Hirai (2015)	[19]	CBDCA-PTX	NA	NA	39	35.9%
Platinum-doublet other chemotherapy						
Okuma (2011)	[20]	CDDP-irinotecan	NA	NA	9	55.6%
Luo (2016)	[21]	CDDP-gemcitabine	NA	NA	13	61.5%

Abbreviations: Ref., reference; TC, thymic carcinoma; ORR, overall response rate; *N*, number of patients; CAP, cisplatin-adriamycin-cyclophosphamide; ADOC, cisplatin-doxorubicin-cyclophosphamide-vincristine; CODE, cisplatin-vincristine-doxorubicin-etoposide; VP16, etoposide; IFO, ifosfamide; DTX, docetaxel; CDDP, cisplatin; PTX, paclitaxel; CBDCA, carboplatin; NA, not applicable.

**Table 2 cancers-13-05441-t002:** Previous reports for S-1 in patients with previously treated thymic carcinoma.

First Author (Year)	Ref.	*N*	Study Design	ORR (%)	mPFS (Months)	mOS (Months)
Tsukita (2020)	[22]	20	Prospective study (Phase II)	25%	5.4	22.7
Okuma (2020)	[23]	26	Prospective study (Phase II)	30.8%	4.3	27.4
Okuma (2016)	[24]	14	Retrospective	42.6%	8.1	30.0
Wang (2019)	[25]	44	Retrospective	30%	6.0	15.0
Hirai (2014)	[26]	8	Retrospective	50%	6.0	13.5
Tateishi (2020)	[27]	18	Retrospective	38.9%	8.3	21.4

Abbreviations: Ref., reference; *N*, number of patients; ORR, overall response rate; mPFS, median progression free survival; mOS, median overall survival.

**Table 3 cancers-13-05441-t003:** Previous reports for amrubicin in patients with previously treated thymic carcinoma.

First Author (Year)	Ref.	*N*	Study Design	ORR (%)	mPFS (Months)	mOS (Months)
Amrubicin monotherapy						
Hellar (2019)	[37]	19	Prospective study (Phase II)	11%	7.3	18
Hirai (2013)	[38]	9	Retrospective	44.4%	4.9	6.4
Tateishi (2020)	[27]	9	Retrospective	14.3%	2.9	40.4
Amrubicin plus carbopatin						
Inoue (2014)	[39]	33	Prospective study (Phase II)	30%	7.6	27.3
Amrubicin plus cisplatin or nedaplatin						
Koizumi (2010)	[40]	6	Retrospective	33.3%	NA	NA

Abbreviations: Ref., reference; *N*, number of patients; ORR, overall response rate; mPFS, median progression free survival; mOS, median overall survival; NA, not applicable.

**Table 4 cancers-13-05441-t004:** Previous reports for other cytotoxic agents in patients with previous treated thymic carcinoma.

First Author (Year)	Ref.	*N*	Study Design	ORR (%)	mPFS (Months)	mOS (Months)
Pemetrexed monotherapy						
Gbolahan (2018)	[45]	11	Prospective study (Phase II)	9%	2.9	9.8
Liang (2015)	[46]	10	Retrospective	10%	6.5	12.7
Qian (2016)	[47]	14	Retrospective	7%	4.5	28.7
Docetaxel based chemotherapy						
Watanabe (2015)	[48]	13	Retrospective	31%	5.5	24.0
Tateishi (2020)	[27]	13	Retrospective	0.0%	2.3	21.7
Song (2014)	[49]	15	Retrospective	26.7%	4.0	22.0
Carboplatin plus Paclitaxel						
Song (2014)	[50]	12	Retrospective	25%	3.5	24.0
Tateishi (2020)	[27]	60	Retrospective	21.2%	6.9	25.3
Komatsu (2006)	[51]	3	Retrospective	66.6%	NA	NA
Gemcitabine monotherapy						
Tateishi (2020)	[27]	8	Retrospective	28.6%	NA	31.8

Abbreviations: Ref., reference; *N*, number of patients; ORR, overall response rate; mPFS, median progression free survival; mOS, median overall survival.

**Table 5 cancers-13-05441-t005:** Previous reports for molecular targeting agents in patients with previous treated thymic carcinoma.

First Author (Year)	Ref.	*N*	Study Design	ORR (%)	mPFS (Months)	mOS (MONTHS)
Everolimus						
Zucali (2017)	[54]	19	Prospective study (Phase II)	16.7%	5.6	14.7
Hellyer (2020)	[55]	3	Retrospective	NA	NA	5.3
Sunitinib						
Thomas (2015)	[56]	25	Prospective study (Phase II)	26%	7.2	NR
Remon (2016)	[57]	20	Retrospective	20%	3.3	12.3
Lenvatinib						
Sato (2020)	[58]	42	Prospective study (Phase II)	38%	9.3	NR

Abbreviations: Ref., reference; *N*, number of patients; ORR, overall response rate; mPFS, median progression free survival; mOS, median overall survival; NR, not reached; NA, not available.

**Table 6 cancers-13-05441-t006:** Previous reports for PD-1 blockade in patients with previous treated thymic carcinoma.

First Author (Year)	Ref.	*N*	Study Design	ORR (%)	mPFS (Months)	mOS (Months)
Cho (2019)	[61]	26	Phase II	19.2%	6.1	14.5
Giaccone (2018)	[62]	40	Phase II	22.5%	4.2	24.9
Katsuya (2019)	[63]	15	Phase II	0%	3.8	14.1

Abbreviations: Ref., reference; *N*, number of patients; ORR, overall response rate; mPFS, median progression free survival; mOS, median overall survival.

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
