# Peer review of "Salvage Chemotherapy in Patients with Previously Treated Thymic Carcinoma"

_cancers, 2021, doi:10.3390/cancers13215441_

Round 1

Reviewer 1 Report

The Authors provided an exhaustive and well written review of the salvage chemotherapy in Thymic epithelial tumors.

Very few suggestiosn regarding the english language or to editing mistakes:

In pg. 1, line 32-33: In patients with metastatic or advanced TC, systemic chemotherapy is widely  chosen to decrease tumor growth and metastasis but is frequently resistant to chemotherapy.....

I WOULD ASK TO WRITE: ......BUT THE TUMOR IS FREQUENTLY RESISTANT TO......(the subject is lacking)

In pg 3, linie 108-109, I would ask to write:   "Due to their rarity,"....  instead of:

Due to rare neoplasms, there are limited reports on salvage chemotherapy for patients with previously treated advanced TC 

In pg 12, linie 454, reference 41:  please change thmic in:  thymic

Reviewer 2 Report

Too much extenive on "Asian limited" data, it misses standard CT approaches and relevant / new targets pathway (ie: KIT, CDK). Also the difference in management between thymoma and TC is confused

As example, see below

Introduction

Line 36: ..there is no established evidence that savage CT..” should be reformulated, being further lines CT usually given with high rate of DCR. Ie: despite no formal evidence of survival benefits with salvage CT, it use I widely supported by clinical practice and high DCR ..”

Line 41: anthracycline are a key drug for thymoma, not the same for TC, this should be stressed. Please add as support meta-analysis

Not reported important issues chemotherapy

Capecitabine and gemcitabine (Palmieri G, Future Oncology 2014)

Oral etoposide (Bluthagen et al Laung Cancer 2016)

Ifosfamide (Highley et al JCO 1999)

Not reported important issues target therapy

CDK inhibitors (ie Milciclib, besse et al, JCO 2018)

KIT inhibitors in KIT mutated TC (series, imatinib and dasatinib)

Other

Line 79: “ … ADOC or CAP has been clarified as a toxic regimen… the useful frequency of these heavy regimen gradually decrease” . Actually, CAP is a standard regimen, widely used and really feasible. Its use is standard, at least in Europe for Thymoma

Too much text is devoted to S1. This drugs has been studied and itsr results have been reported only in Asian series, and it has not been tested in Caucasian. It is not used outside of Asia.

Also ambrubicina is not a key drug in Europe and USA and it has been too much discussed in this article

Pemetrexed: Xinyu Qian (oncotargets and therapy 2014) is missing

Reviewer 3 Report

This manuscript is a nice review of systemic therapies for thymic carcinoma.

Well written and extensive, up to date;

Reviewer 4 Report

very good paper

I will suggest 3 points:

  • mention NIVOTHYM (ESMO 2021) 
  • mention study from PALMIERI CAPGEM 
  • make a table with immuno-check points inhibitors explored in this setting even it has aleready descripted by the authors in another rewview (réf 3)

Round 2

Reviewer 2 Report

nothig to be added